# Automated Laryngeal Invasion Detector of Boluses in Videofluoroscopic Swallowing Study Videos Using Action Recognition-Based Networks

**DOI:** 10.3390/diagnostics14131444

**Published:** 2024-07-06

**Authors:** Kihwan Nam, Changyeol Lee, Taeheon Lee, Munseop Shin, Bo Hae Kim, Jin-Woo Park

**Affiliations:** 1Graduate School of Management of Technology, Korea University, Seoul 02841, Republic of Korea; namkh@korea.ac.kr; 2AimedAI, Seoul 06150, Republic of Korea; charlie@aimed.ai; 3Department of Physical Medicine and Rehabilitation, Dongguk University Ilsan Hospital, College of Medicine, 27 Dongguk-ro, Ilsandong-gu, Goyang 10326, Republic of Korea; taeheon320@naver.com (T.L.); shinms306@gmail.com (M.S.); 4Department of Otorhinolaryngology-Head and Neck Surgery, Dongguk University Ilsan Hospital, College of Medicine, 27 Dongguk-ro, Ilsandong-gu, Goyang 10326, Republic of Korea

**Keywords:** artificial intelligence, deep learning, deglutition disorders, fluoroscopy

## Abstract

We aimed to develop an automated detector that determines laryngeal invasion during swallowing. Laryngeal invasion, which causes significant clinical problems, is defined as two or more points on the penetration–aspiration scale (PAS). We applied two three-dimensional (3D) stream networks for action recognition in videofluoroscopic swallowing study (VFSS) videos. To detect laryngeal invasion (PAS 2 or higher scores) in VFSS videos, we employed two 3D stream networks for action recognition. To establish the robustness of our model, we compared its performance with those of various current image classification-based architectures. The proposed model achieved an accuracy of 92.10%. Precision, recall, and F1 scores for detecting laryngeal invasion (≥PAS 2) in VFSS videos were 0.9470 each. The accuracy of our model in identifying laryngeal invasion surpassed that of other updated image classification models (60.58% for ResNet101, 60.19% for Swin-Transformer, 63.33% for EfficientNet-B2, and 31.17% for HRNet-W32). Our model is the first automated detector of laryngeal invasion in VFSS videos based on video action recognition networks. Considering its high and balanced performance, it may serve as an effective screening tool before clinicians review VFSS videos, ultimately reducing the burden on clinicians.

## 1. Introduction

Swallowing is successfully accomplished through the sequential and harmonious movement of the upper digestive tract structures [1]. Functional loss or anatomical deformities of these structures, which play a pivotal role in the swallowing process, can impair bolus transition, resulting in conditions such as aspiration pneumonia or malnutrition [2]. Videofluoroscopic swallowing study (VFSS) is the most valuable diagnostic test for dysphagia, elucidating the compromised swallowing mechanism by capturing serial images while the patient ingests a fluorescent bolus [3]. Converting the images acquired during VFSS into a chronological video format facilitates a visually intuitive and detailed evaluation of the dynamic movement of the swallowing structures and their correlation with bolus transition [4]. Consequently, clinicians primarily rely on reviewing VFSS videos. As physicians must assess the success of the intricate swallowing process through reconstructed VFSS videos, which document the entire process within a few seconds, this review demands a significant investment of time and experience from physicians. Moreover, issues related to intra- and inter-rater reliabilities need to be addressed when scrutinizing VFSS videos [5]. 

Artificial intelligence (AI) based on deep learning has been expanding its applications in various medical fields to enhance the diagnosis and treatment of diseases, alleviate the burden on physicians, and improve the reliability of image review [6,7]. Among the different deep learning models for detecting abnormalities in medical images, image-based analysis, which typically utilizes a two-dimensional (2D) convolutional neural network (CNN), is the most widely employed [8]. A 2D CNN can identify image characteristics by constructing multiple convolution and subsampling layers. It has demonstrated excellent performance in classifying and identifying abnormalities in medical images, despite the relatively limited volume of learning data available in the medical field compared with that in the information technology industry [8,9]. In a previous study, an image-based network exhibited a highly proficient ability to detect aspiration on VFSS [10]; however, image-based networks focus on identifying the spatial characteristics of an image and often overlook temporal information (ordered frame sequences) [11,12]. Therefore, a new deep learning network should be capable of analyzing data, considering both spatial and temporal information, as required in VFSS [13,14]. 

The most crucial aspect of accurate decision-making by AI using VFSS videos is the efficient extraction and learning of meaningful bolus movements from a few VFSS videos [15]. Action recognition is a specialized model for video analysis that utilizes both spatial and temporal information [16]. Additionally, action recognition can produce more meaningful results in video analysis than conventional image-based analysis because it efficiently analyzes and predicts the characteristics of VFSS videos. Therefore, we aimed to develop an automated detector that determines laryngeal invasion during swallowing by utilizing two or more scores on the penetration–aspiration scale (PAS). This scale poses significant clinical problems that we aimed to address in VFSS videos by employing two 3D stream networks for action recognition. 

## 2. Materials and Methods

### 2.1. Video Fluoroscopic Swallowing Study 

All the VFSS procedures were supervised by a single experienced physician in the Department of Rehabilitation Medicine (J.W.P.). VFSS images were acquired while the patients swallowed a fluorescent bolus mixed with liquid barium and water in an upright position, positioned 1.5 m away from the X-ray tube (Sonialvision-100; Shimadzu Corporation, Kyoto, Japan). Each participant completed two swallows with a 5 mL bolus administered using a syringe. Lateral-view images were captured at 30 frames per second with a frame size of 1021 × 1021 pixels and digitally stored in a digital picture archive and communication system [17]. 

### 2.2. Dataset

We consecutively extracted 1300 VFSS videos from our institution’s database of patients complaining of dysphagia between January 2010 and April 2022. The study protocol was approved by our institutional review board (IRB) [IRB no. 2022-05-007-001]. The dataset included VFSS videos of appropriate image quality that recorded at least one task of the entire swallowing process, specifically focusing on the fluidic bolus, which is the most sensitive to laryngeal invasion. We included all recorded videos of liquid swallowing in adults aged 18 years, regardless of illness. After excluding videos with poor quality or those that did not capture the entire swallowing process, 1023 videos were consecutively included.

Two specialized reviewers determined the PAS scores of the included VFSS videos. Any disagreements were resolved by a third reviewer. Laryngeal invasion during swallowing on VFSS videos was defined as PAS 2 or higher [18]. PAS 1 was assigned to 266 videos (26.0%), and 757 videos (74.0%) were assigned two or more points. The dataset was randomly divided into two groups: a training set (821 videos) and a testing set (202 videos). The testing set comprised 51 PAS 1 videos and 151 laryngeal invasion videos (PAS 2 or higher), serving to evaluate the performance of our AI model.

### 2.3. Deep Learning Architecture for Action Recognition 

This study mainly focused on the application of action recognition techniques to identify laryngeal invasion (PAS 2 or higher scores) in VFSS videos. A video classification model, employing a 2D CNN that disregards spatial factors, extracted features from individual frames using an image classification network. These features were then amalgamated to obtain the results (Figure 1A). However, because spatial factors were not considered, this approach exhibited suboptimal performance in intricate videos such as VFSS [19]. 

The fundamental principle of the action recognition model involves recognizing keyframe information in a multiframe sequence and classifying it based on the selected keyframe data. Four dimensional factors (two spatial dimensions, height and width, one temporal dimension, and one channel dimension) traversed the model, facilitating the learning of various temporal interactions between adjacent frames (Figure 1B) [19]. The total number of frames in the video, denoted as “n” for each VFSS test image, and the specific starting and ending points relevant to the VFSS model analysis varied. To ensure an efficient analysis, we defined valid starting and ending points within the ‘n’ frames and interpolated the selected frames using the nearest-neighbor method [20]. Consequently, “k” fixed-size frames were generated and employed as inputs for the action recognition network. To develop a model optimized for video-based action recognition, accuracy was enhanced and computational costs were reduced using a 3D group convolution network [21]. Model learning involved the incorporation of group and depth-wise convolutions. Group convolution was partitioned into channel interaction and spatial–temporal interaction to augment accuracy and introduce a regularization effect to mitigate overfitting [22]. Furthermore, depth-wise convolution was employed to reduce computational costs [22]. As the video underwent convolution, it was segmented into R, G, and B channels, enabling the model to concurrently learn from these three channels, thereby reducing the computational costs by a factor of three.

### 2.4. Application of Architecture for Action Recognition on VFSS 

The videos stored in our institutional database were converted into individual frame images. To enhance image comprehension, we equalized the brightness of all the images. The region of interest (ROI) on each image was divided into three areas where bolus transition occurred during swallowing: the pharynx, larynx, and vocal folds. Manual annotation was performed for each frame image in these areas. After selecting the frames indicating the initiation and completion of swallowing, we sorted the annotated images in the ROI according to time sequence. These images were initially interpolated using the nearest-neighbor method and subsequently used to train a deep learning model. We concatenated all the learned features extracted from the annotated images, followed by the extraction of the final features [23]. Finally, we analyzed the detector performance using a test set to identify abnormalities in the VFSS video (Figure 2). To establish the model’s robustness, we compared its performance with that of various up-to-date image classification-based architectures (resnet101, swin-transformer, efficientnet-β2, and HRnet-w32).

## 3. Results

### Performance 

The accuracy of our model for classifying laryngeal invasion was 92.10% (Table 1). The specific accuracy of our model was 84.31% (43/51 videos) for the prediction of PAS 1 and 94.70% (143/151 videos) for PAS 2 or higher (Figure 3A). The receiver operating characteristic (ROC) curve showed an area under the curve of 0.88 (Figure 3B). The precision, recall, and F1 scores for detecting laryngeal invasion (PAS 2 or higher) in the VFFS video were 0.9470, 0.9470, and 0.9470, respectively (Table 1). As the precision, recall, and F1 score for detecting PAS 1 scores were 0.8431, 0.8431, and 0.8431, respectively (Table 1), our model performed better in detecting laryngeal invasion compared to determining the absence of laryngeal invasion on the VFSS video. This result may be attributed to the fact that the number of laryngeal invasion videos used for training the model was greater than the number of PAS videos. When using the proposed model to classify the absence of laryngeal invasion (PAS 1) and the presence of laryngeal invasion (PAS 2 or higher), the most common misclassification was PAS 2, classified as the absence of laryngeal invasion (PAS 1). The remaining PAS 3 to PAS 8 showed solid results in terms of prediction accuracy.

The accuracy of our model in identifying laryngeal invasion was higher than that of other up-to-date image classification models (60.58% for ResNet101, 60.19% for Swin-Transformer, 63.33% for EfficientNet-B2, and 31.17% for HRNet-W32) (Table 2).

## 4. Discussion

To determine laryngeal invasion in VFSS videos, we developed an automated detector based on two 3D stream networks for action recognition. The classifier was designed using 3D convolutions to learn both spatial features and movements based on the time sequence of the bolus [24,25]. Our detector for VFSS videos demonstrated an accuracy of 92.1%, surpassing that of various state-of-the-art image classification-based deep learning networks.

Action recognition in videos using deep learning, applied for various purposes such as ensuring public safety, preventing crimes, and enhancing the effective motion of athletes, is a technique that recognizes or classifies actions within the object of interest [26,27]. Advances in action recognition and handling of spatial–temporal information have been delayed compared with those in image analysis models owing to the lack of large-scale datasets, high computational cost, and less attention to temporal modeling [28]. However, video action recognition has recently progressed with the introduction of various architectures that can reduce computational costs, while effectively learning temporal and spatial information from videos [28]. VFSS contains 3D information on bolus transitions with height, width, and time sequence. Clinicians can determine abnormalities in VFSS by considering the bolus location and transition [4]. Therefore, spatial–temporal models for action recognition are suitable for application in VFSS deep learning. Additionally, VFSS videos composed of serial images can be used to capture the swallowing process in a short time. This is also an advantageous characteristic of VFSS videos when applying a spatial–temporal model for action recognition from the perspectives of computational cost and clinical application. We defined the bolus transition according to the time in the VFSS videos as an action of interest. Our model is composed of two main 3D convolutions, one for spatial–temporal interaction and the other for channel interaction, to handle both spatial and temporal information in the VFSS video. Moreover, convolution separation enables various calculations by simultaneous parallel processing and reduces the number of parameters used in each convolution, resulting in a reduced computational cost. This network is generally called the “two-stream 3D network for action recognition” [29]. The next important step in the final prediction was to fuse the two separately trained 3D streams by averaging the results of both convolution predictions. In our model, the final prediction was made based on the average output of the two streams through fusion using the concatenation of all learned features from the annotated images [23]. 

An automated detector was recently developed to determine the presence of aspiration in VFSS by utilizing a 2D CNN for detection [10]. This classifier focused on a specific pathological event and demonstrated an accuracy of 93.2%, with 91.2% recall and 88.1% precision in detecting aspiration during swallowing. The remarkable performance of previous studies may be attributed to the extraction of image characteristics through multiple convolutions with various kernels specialized in detecting single pathological events such as aspiration [10]. Classifying VFSS videos into PAS 1 and PAS 2 or higher groups using a conventional CNN necessitates the definition and learning of all the diverse pathological events observed in VFSS, incurring excessive work and costs. We closely simulated the actual VFSS review process for clinicians using our deep learning model for bolus action recognition. This model exhibited high and balanced performance with 92.10% accuracy, 94.7% precision, 9474% recall, and a 0.947 F1 score for detecting laryngeal invasion during swallowing in VFSS videos. In contrast to the 2D CNN, our action recognition model learns bolus characteristics by incorporating channel interaction and bolus transition, in addition to spatial–temporal interaction, to enhance accuracy and introduce a regularization effect. This is achieved without the need to define and train various types of pathological events in VFSS [30]. This aspect is a crucial control point that can contribute to the high and balanced performance of our classifier, while minimizing the training burden for the deep learning model.

Although various advanced techniques for action recognition have been employed to address the limited training data, only approximately 1000 VFSS videos were used for both model training and validation in this study. The scarcity of data poses a significant issue and is a common challenge in the development of AI models for medical applications. Moreover, external validation using VFSS data from other institutions was not performed during the model development phase. Discrepancies in image quality and VFSS protocols can affect the effectiveness of deep learning in AI development. The primary limitations of this study include the insufficient volume of data and the absence of external validation. These limitations could be mitigated by incorporating additional external data and expanding the dataset.

To the best of our knowledge, our model is the first automated detection of laryngeal invasion in VFSS videos that utilizes video action recognition networks. Owing to its demonstrated high and balanced performance, it may serve as an effective screening tool before clinicians review VFSS videos, potentially reducing their burden.

## Figures and Tables

**Figure 1 diagnostics-14-01444-f001:**
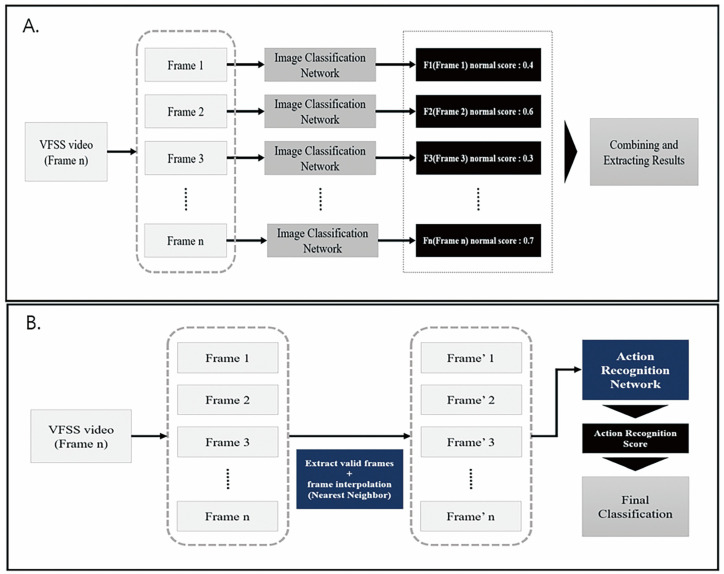
Presentations of network architecture for analyzing medical data: (**A**) imaging-based deep learning models primarily determine image normality based on extracted characteristics from the dataset without considering temporal information from serial images; (**B**) action recognition networks predict the abnormality of VFSS videos by incorporating both spatial and temporal information. VFSS—videofluoroscopic swallowing study.

**Figure 2 diagnostics-14-01444-f002:**
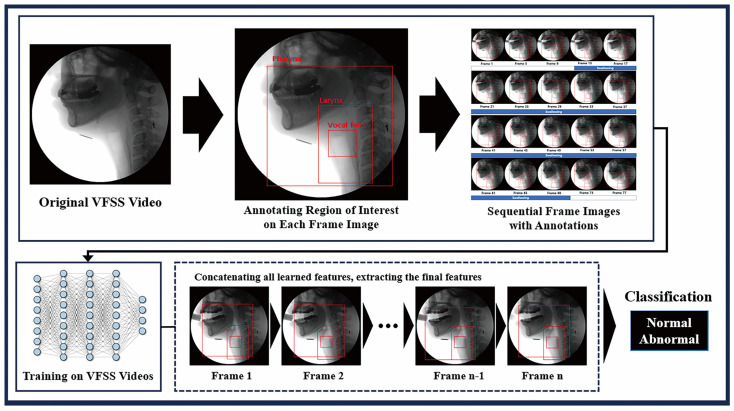
Process for training the deep learning model using VFSS videos. The training of a two-stream 3D network is initiated by annotating the region of interest in each frame image. After sorting annotated images according to the time sequence, they are utilized for deep learning model training. All learned features from annotated images are concatenated, and final features are then extracted. VFSS—videofluoroscopic swallowing study.

**Figure 3 diagnostics-14-01444-f003:**
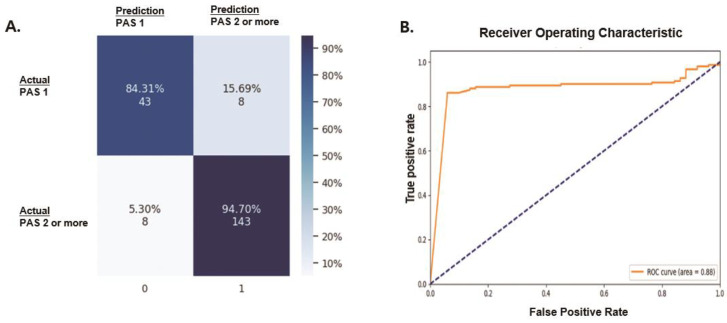
Performance of our action recognition model for detecting laryngeal invasion in VFSS videos: (**A**) confusion matrix; (**B)** receiver operating characteristic curve for detecting laryngeal invasion using VFSS videos. VFSS—videofluoroscopic swallowing study; PAS—penetration–aspiration scale.

**Table 1 diagnostics-14-01444-t001:** Precision, recall, and F1 scores per video determining the normality of videofluoroscopic swallowing study videos.

Classification	Precision	Recall	F1 Score
Absence of laryngeal invasion (PAS 1)	0.8431	0.8431	0.8431
Presence of laryngeal invasion (PAS 2 or higher)	0.9470	0.9470	0.9470

PAS—penetration–aspiration scale.

**Table 2 diagnostics-14-01444-t002:** Comparison of accuracy between our model and other up-to-date image classification models.

Type	Model	Valid Accuracy
Image	resnet101	60.39% (122/202)
	swin-transformer	60.19% (125/202)
	efficientnet-b2	63.36% (128/202)
	HRnet-w32	61.38% (124/202)
Video	Our model	92.10% (186/202)

## Data Availability

The data presented in this study are available on request from the corresponding author. The data are not publicly available due to ethical issue.

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
