# Peer review of "Automated Laryngeal Invasion Detector of Boluses in Videofluoroscopic Swallowing Study Videos Using Action Recognition-Based Networks"

_diagnostics, 2024, doi:10.3390/diagnostics14131444_

Round 1
Reviewer 1 Report
Comments and Suggestions for Authors
Dear Editor
Thanks a lot for your hard work. I read this article with interest. However, I have some concerns.
Kindly incorporate the responses within the manuscript to augment its overall quality.
The authors should clearly determine the criteria for including and excluding from the study group.
They should improve a stastic study and determine which data is constant and which are not, and not just based on a percentage.
In the results, you should show more clearly in the form of numbers (n)
Comments on the Quality of English Language
There are many spelling and language mistakes and the manuscript needs to be corrected by a native English speaker.
Author Response
Comments 1: The authors should clearly determine the criteria for including and excluding from the study group.
Response 1: Thank for your meaningful review and pointing this out. We agree with this comment. Therefore, we additionally described inclusion criteria in the method section (page 2, line 92-94) as follows:
We included all recorded videos of liquid swallowing in adults aged 18 years regardless of illness. After excluding videos with poor quality or those that did not capture the entire swallowing process
Comments 2: They should improve a stastic study and determine which data is constant and which are not, and not just based on a percentage. In the results, you should show more clearly in the form of numbers (n).
Response 2: Thank you for your meaningful comment and pointing this out. We agree with this comment. As you mentioned, the entire number (n) was applied to the prediction results (page 6, Table 2, and line 184). In addition, proposed model was tested several times to extract the average accuracy, and based on the final model results, the total number and correct number were assessed and reflected. We also added explanations on which data showed high accuracy and which data did not produce good results (page 5, line 167-170).
When using the proposed model to classify the absence of laryngeal invasion (PAS1) and the presence of laryngeal invasion (PAS2 or higher), the most common misclassification was PAS2 classified as the absence of laryngeal invasion (PAS1). The remaining PAS3 to PAS8 showed solid results in terms of prediction accuracy.
If this is different from what you intended, please let us know and we will edit it again.
Comments 3. Extensive editing of English language required
Response 3. Thank you for pointing this out. We additionally edited English language in the revised manuscript to professional English editing company.
Reviewer 2 Report
Comments and Suggestions for Authors
1. The question is original and well-defined. The results provide an advancement of the current knowledge.
2. The main question is addressed by the research. It is quite novel.
3. The cited references mostly recent publications (within the last
5 years) and relevant. The number of citations is satisfying.
4. The tables and the whole presentation are also satisfying. I have no comments on that.
5. The manuscript scientifically sounds and the experimental design os appropriate to test the hypothesis.
6. The manuscript’s results reproducible based on the details given
in the methods section.
Author Response
Reviewer 2.
Comments: 1. The question is original and well-defined. The results provide an advancement of the current knowledge.
2. The main question is addressed by the research. It is quite novel.
3. The cited references mostly recent publications (within the last 5 years) and relevant. The number of citations is satisfying.
4. The tables and the whole presentation are also satisfying. I have no comments on that.
5. The manuscript scientifically sounds and the experimental design os appropriate to test the hypothesis.
6. The manuscript’s results reproducible based on the details given in the methods section.
Response: Thank you for positive review result about our manuscript.
Round 2
Reviewer 1 Report
Comments and Suggestions for Authors
I accept in present form